# Micro-PINGUIN: Microtiter plate-based ice nucleation detection in gallium with an infrared camera

Corina Wieber[1,2], Mads Rosenhøj Jeppesen[3], Kai Finster[1,2,4], Claus Melvad[2,3,5], Tina Šantl-Temkiv[1,2,4,5]

[1]Department of Biology, Microbiology, Aarhus University, Aarhus, 8000, Denmark
[2]iCLIMATE Aarhus University Interdisciplinary Centre for Climate Change, Roskilde, 4000, Denmark
[3]Department of Mechanical and Production Engineering, Aarhus University, Aarhus, 8000, Denmark
[4]Stellar Astrophysics Centre, Department of Physics and Astronomy, Aarhus University, Aarhus, 8000, Denmark
[5]Arctic Research Centre, Aarhus University, Aarhus, 8000, Denmark

*Correspondence to*: Tina Šantl-Temkiv (temkiv@bio.au.dk)

**Abstract.**

Ice nucleation particles play a crucial role in atmospheric processes e.g., they can trigger ice formation in clouds and thus influence their lifetime and optical properties. The quantification and characterization of these particles require reliable and precise measurement techniques. In this publication, we present a novel droplet freezing instrument to measure immersion freezing of biotic and abiotic ice nucleating particles within the temperature range of 0 °C to -25 °C. Immersion freezing of
the samples is investigated using 384-well PCR plates with a sample volume of 30 µl. Nucleation events are detected with high precision using a thermal camera that records the increase in infrared emission due to the latent heat release. To maximize the thermal contact between the PCR plate and the surrounding cooling unit, we use a gallium bath as a mount for the PCR plate. The instrument was validated relative to a calibrated temperature standard, as well as through reproducibility measurements employing the same suspension. We find that the combination of good thermal connectivity and precise
temperature recording enables accurate (± 0.81 °C at -10 °C) and reproducible (± 0.20 °C) detection of the nucleation temperatures. Consequently, the results that will be produced using the micro-PINGUIN are of good quality and the instrument can be used to study immersion freezing of various ice nucleating particles.

For comparison with already existing instruments, Snomax® (hereafter Snomax) and Illite NX suspensions are measured with the new ice nucleation instrument, "micro-PINGUIN". Further, we investigated the reproducibility of experiments using
Snomax suspensions and found poor reproducibility when suspensions were prepared freshly even if the same batch of Snomax is used. This could be attributed to substrate heterogeneity, aging effects, and dilution errors. The reproducibility of the measurements is greatly improved for Snomax suspensions that are prepared in advance and stored frozen in aliquots. Thus, we suggest the use of suspensions frozen in aliquots for further reproducibility measurements and intercomparison studies.

# 1    Introduction

Clouds play an important role in the Earth's radiative balance and climate. Cloud properties such as reflectivity and lifetime are to a large extent determined by the properties of atmospheric aerosols. Precipitation from mixed-phase and cold clouds that is initiated via the formation of ice particles significantly contributes to the global water cycle (Mülmenstädt et al., 2015). While homogenous freezing of cloud droplets only takes place in cold clouds (< -37 °C), ice nucleating particles (INPs) are required to initiate freezing in mixed-phase clouds (between 0 °C and -37 °C) (Murray et al., 2012). So far only biological

INPs were shown to initiate ice formation at temperatures higher than -15 °C at atmospherically relevant concentrations (Murray et al., 2012). This can result in a large fraction (up to 83 % in the Arctic) of ice-containing clouds within this temperature range (Griesche et al., 2021).

To study the nature and concentration of INPs, a variety of droplet freezing techniques has been developed. These droplet freezing techniques essentially differ in 1. the sample volume, 2. the number of droplets investigated per run, 3. the cooling

system, and 4. the method used for the detection of nucleation events and temperatures. An overview of various droplet freezing techniques is given by Miller et al. (2021). The sample volume used to investigate the ice nucleation efficiency has a big impact on the type of INPs that can be detected. Instruments with nanolitre and picolitre volumes are primarily used to investigate abundant INPs active at low temperatures (Peckhaus et al., 2016; Reicher et al., 2018; Chen et al., 2018; Budke and Koop, 2015). For these small volumes, the probability of contamination in the negative control is lower, which leads to low freezing

temperatures, typically between -30 °C to -37 °C. Thus, activity can be investigated for samples that nucleate close to the temperatures at which homogenous freezing is initiated. However, low-volume instruments require a high concentration of INPs in the sample to be detected. Therefore, small volume instruments are less suitable for the analysis of high-temperature INPs, which are often present at low concentrations. In contrast, instruments with larger sample volumes, allow the study of rare INPs. However, with larger volumes the presence of impurities in the water control is becoming more likely, leading to a

higher background freezing temperature. Consequently, it is challenging to study INPs that are active at low temperatures with large volume droplet freezing techniques as the freezing curves at lower temperatures start to overlap with the curves of the pure water background. For instruments with a sample volume of 50 µl, freezing events in the negative control, are reported between -20 °C and -27 °C (Schiebel, 2017; Harrison et al., 2018; Miller et al., 2021; Beall et al., 2017; Barry et al., 2021; David et al., 2019; Gute and Abbatt, 2020). To obtain a high quality of the freezing spectra, a sufficient number of droplets

must be analyzed. This can be achieved by investigating a large number of droplets per run or by repeated experiments of the same substance. A recent modelling study has shown that a small number of droplets (<100) leads to poor statistics and can cause misrepresentation of the underlying INP distribution (De Almeida Ribeiro et al., 2023). Furthermore, droplet freezing techniques differ in the method they use to cool the samples. The cooling system for the droplet freezing techniques is often composed of a liquid cooling bath (Gute and Abbatt, 2020; Chen et al., 2018; David et al., 2019; Miller et al., 2021) or

thermostats circulating a cooling liquid through the cooling block (Beall et al., 2017; Schiebel, 2017; Kunert et al., 2018).

Other instruments are based on a cold stage cooled by liquid nitrogen (Peckhaus et al., 2016), Peltier elements (Budke and Koop, 2015; Chen et al., 2018) or a Stirling engine-based cryocooler (Harrison et al., 2018; Tobo et al., 2019). Another difference between the droplet freezing techniques is the way they determine the freezing of the samples. To detect the freezing events, several instruments use an optical camera, combined with a temperature sensor to measure the freezing temperatures.

As the temperature is measured only at positions where a temperature sensor is placed, gradients within the instrument lead to a reduced accuracy of the detected freezing points. To minimize these gradients, good thermal conductivity of the materials used between the cooling unit and the sample is of great importance. Further, the detection of the freezing temperatures with an optical camera is usually not based on the detection of the ice nucleation event but is instead based on the change in optical properties such as brightness of the sample during the process of the whole droplet freezing. As a result, the detection of the

nucleation temperature based on changing optical properties is challenged by the fact that the total freezing time can take up to several minutes, in particular at larger volumes and temperatures relevant for biogenic INPs. Consequently, it is difficult to determine the exact starting point of ice nucleation using an optical camera. Harrison et al. (2018), for example, reported a freezing time of 100 seconds for a 50 µl droplet freezing at -12 °C. The delay between the nucleation event and the freezing of the whole droplet can thus result in an error of >1.5 °C at -12 °C, assuming a cooling rate of 1 °C min$^{-1}$ and larger errors are

expected at higher nucleation temperatures that are relevant for biogenic INP.

Despite the differences in the design of various droplet freezing techniques, the results they produce have to be comparable across the instruments. Therefore, there has been carried out a series of intercomparison studies of various instruments using compounds such as Snomax (Wex et al., 2015) and Illite NX (Hiranuma et al., 2015). Snomax is a commercially available product that consists of freeze-dried cell material of the ice nucleation active bacterium *Pseudomonas syringae* with freezing

temperatures as high as -2 °C and Illite NX is a mineral mix that contains Illite, Kaolinite, Quartz, Carbonate and Feldspar ice nucleation active at temperatures below -11 °C. Both substances were found to be suitable for intercomparison studies when taken from the same batch, used at similar concentrations, and stored only short-term. However, Polen et al. (2016) observed that the ice nucleation ability of very active proteins in Snomax powder changes over time during storage in the freezer, which they suggest is due to aging, thus emphasising necessity of short storage times. The fact that the ice nucleation capacity of

Snomax is unstable over time causes problems in intercomparison studies and leads to deviation in the measured ice nucleation activity that can span over several orders of magnitude.

In this publication, we present a novel ice nucleation instrument for **micro**titer **p**late-based **i**ce **n**ucleation detection in **g**alli**u**m with an **in**frared camera (micro-PINGUIN). A high accuracy is achieved by combining good thermal contact between the sample and the surrounding cooling unit with the detection of freezing events by an infrared camera. The working principle

and the validation of micro-PINGUIN are described in the following sections. Furthermore, we address the challenges due to inhomogeneities of the product and due to aging effects and propose a possible solution for using Snomax as a suspension for intercomparison studies and reproducibility measurements.

## 2    Description of the instrument

Figure 1 shows a schematic drawing of the micro-PINGUIN instrument. It is composed of three main parts: the cooling unit,
the camera tower with an infrared camera for the detection of freezing events and the electronic components to control the
instrument and measure the temperatures. A photograph of the instrument is provided in the Supplementary S1.

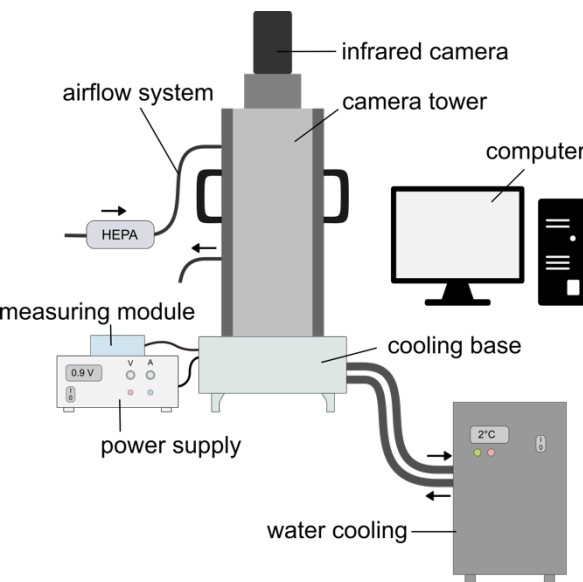

**Figure 1:** Schematic drawing of the micro-PINGUIN instrument.

The cooling base of the micro-PINGUIN instrument (Fig. 2) is built up from several layers. Primarily, two Peltier elements, a
vapor chamber, and a gallium bath are used to achieve good thermal conductivity, reduce horizontal gradients, and optimize
the cooling capacity of the instrument. The temperature and the cooling gradients are controlled by two PID-regulated Peltier
elements (QC-241-1.6-15.0M, QuickCool, Germany). By applying voltage to the Peltier elements, a temperature difference is
achieved between the two sides of the element. A water-cooling bath (Eiszeit 2000, Alpha cool, Germany) is connected to the
cooling unit and circulates precooled water (2 °C) through the water cooler base plates (Cuplex kryos NEXT sTRX4 FC, Aqua
Computer, Germany). Thereby, the heat generated on the lower side of the Peltier elements is removed. The Peltier elements
are positioned within a copper base, to which they are connected by thermal pads. Above the lower copper plate, a vapor
chamber is used to distribute the temperature evenly and thus minimize horizontal temperature gradients within the instrument.
A second copper base is positioned above the vapor chamber. It contains a fix-point cavity for the temperature measurement
with the infrared camera and a Pt100 temperature probe (RTDCAP-100A-2-P098-050-T-40, Omega, Denmark) at the same
position to achieve precise temperature measurements as described in detail in paragraph 2.3. The upper copper base contains
a gallium bath that melts at around 30 °C. A 384-well PCR plate (384 PCR plate full skirt, Sarstedt, Germany) is submerged

in the melted gallium bath that solidifies when cooled to room temperature. Thereby, thermal contact between the PCR plate and the cooling unit is obtained. The freezing events are detected with a thermal camera (FLIR A655sc/25° Lens, Teledyne Flir, US) that is mounted in a black-painted camera tower and positioned above the PCR plate. A continuous flow (10 l min$^{-1}$) of air with a low relative humidity (<10 % RH) is circulated within the camera tower to keep the humidity low and to avoid condensation of water vapor on the PCR plate, which would interfere with the experiments. The instrument is controlled by a custom-made software (Ice Nucleation Controller). Per default, a cooling run is started at 10 °C with a cooling rate of 1 °C min$^{-1}$ until the final temperature of -30 °C is reached.

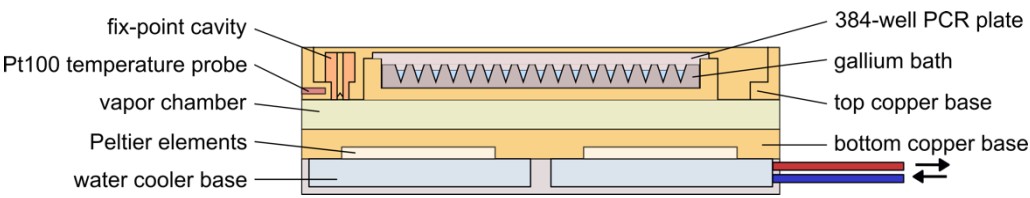

**Figure 2:** Schematic drawing of the cooling base. The red and blue tubes connected to the water cooler base indicate the circulation of cooled water which removes the heat generated by the Peltier elements.

## 2.1    The role of the gallium bath

For an optimal cooling performance of the instruments, materials with high thermal conductivity are used. Aluminum is a commonly used material in PCR plate based instruments (Schiebel, 2017; Kunert et al., 2018; Beall et al., 2017; Hill et al., 2014) as it has a good thermal conductivity and is easy to shape, which makes it suitable for the PCR plates mount. However, it cannot be avoided that a thin layer of air between the PCR plate and the aluminum plate forms and because of the insulating properties of air, the thermal contact between the cooling system and the samples is hampered. To maximize thermal contact with the sample while minimizing the manufacturing process of the PCR mounting plate, we used gallium as a mount for the PCR plate (Supplementary S2). Gallium is a metal with a low melting temperature (29.8 °C) and a high thermal conductivity (29.3–37.7 W m$^{-1}$ K$^{-1}$) (Prokhorenko et al., 2000). In the PINGUIN instrument, we use gallium to connect the 384-well PCR plates to the cooling system. By reversing the polarity on the Peltier elements, the instrument can be heated to 40 °C, causing the gallium to melt. The PCR plate is then inserted into the liquid gallium and the instrument is cooled to 10 °C. During this process, the gallium solidifies while close contact with the PCR plate is created, and any excess air is pushed out. To achieve a uniform contact for the whole plate, a predetermined weight is placed on top of the PCR plate during the solidification of the gallium. After mounting the PCR plate, 30 µl of the samples per well are distributed using an automatic 8-channel pipette (Pipetteman P300, Gilson, US). For future versions of this instrument, we are planning to make small modifications, demonstrating that the use of gallium has further advantages. As heating can have an impact on the ice nucleation activity of the samples, the wells are usually filled with the suspension after the heating-cooling cycle. However, this procedure can be

an additional advantage of using gallium as a mount for the PCR plate. The use of gallium as a heat conductive medium allows
to apply precise heat treatments to the samples by controlling the temperature of the gallium bath. Consequently, previously
measured plates could be remeasured after heating the gallium to the desired temperatures. Heat treatments are a commonly
used method to differentiate between different biogenic and inorganic INPs, assuming that biogenic INPs will decrease in the
ice nucleation activity when heated to sufficiently high temperatures (an overview of studies using heat treatments is given in
Daily et al. (2022)). The approach presented here could substitute traditional heating methods such as ovens or water baths for
treating the samples. This would not only simplify the experimental process but also facilitate accurate and reproducible heat
treatments for ice nucleation activity studies. Further, when using gallium as a mount for the PCR plates the instrument is not
limited to one type of PCR plates. Small modifications to the instrument allow the use of 96-well plates instead of 384-well
plates, thus extending the range of sample volumes and thus INP concentrations that can be investigated.

## 2.2    The airflow system

During initial tests of the micro-PINGUIN instrument, freezing temperatures as high as -13 °C were observed for the negative
control. Similar freezing temperatures were found for MilliQ water, tap water and ultra-pure water for molecular work and
were not affected by filtration or autoclaving of the water, indicating that these high freezing temperatures are not caused by
impurities in the water. During the experiments, condensation of water vapor on the copper base and the PCR plate was
observed. Tests with a flow of compressed air with a low humidity (<10 % RH) passing through the camera tower showed that
condensation was avoided during the experiment and that the freezing temperatures decreased with increasing air flow until
$T_{50}$ temperatures, corresponding to the temperature where 50 % of the droplets are frozen, of around -25 °C are reached
(Fig. 3). These background freezing temperatures are common for freezing experiments with volumes in the microlitre range.
Other instruments using volumes of 50 µl (Schiebel, 2017; Harrison et al., 2018; Miller et al., 2021) reported comparable
frozen fraction curves obtained by measurements of their negative controls. These airflow experiments indicated that at high
humidity conditions, the freezing was caused by condensed water on the plates instead of INPs in the suspension. Thus, we
decided to apply a flow of dry air that is injected at the top part of the camera tower to lower the humidity in the
micro-PINGUIN instrument. Before each run, the camera tower is flushed with a high flow of dry air (20 l min$^{-1}$). The flow is
reduced to 10 l min$^{-1}$ during the measurement to minimize disturbance of the samples and the introduction of warm air. We
measured the relative humidity in the camera tower for this procedure and found that a flow of 10 l min$^{-1}$ is sufficient to
maintain a low relative humidity during the experiment. Further, we have evaluated the sample loss due to evaporation and
found that this factor is negligible as only 0.36% of liquid was lost during an experiment. With this procedure, the $T_{50}$
temperatures of the negative control was usually as low as -25 °C. Also other droplet freezing techniques apply a flow of dry
air or $N_2$ to the instrument to avoid frost formation (Schiebel, 2017; Budke and Koop, 2015).

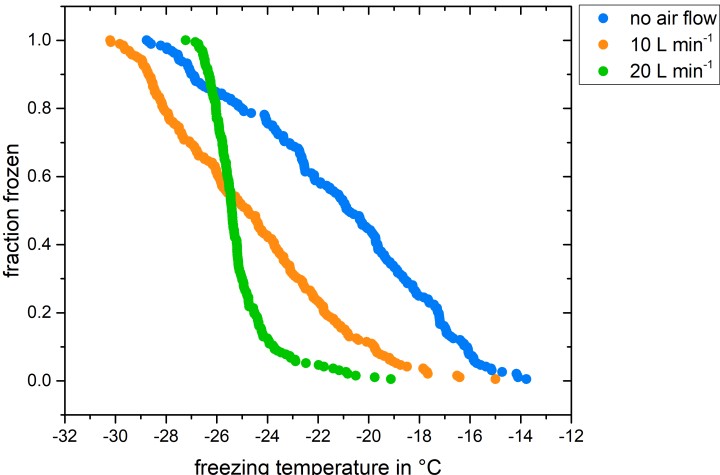

**Figure 3:** Effect of a dry air flow on the freezing behavior of the negative control (MilliQ water). These freezing curves were obtained without flushing the camera tower with dry air prior to the experiment.

## 2.3 Temperature measurement and detection of freezing events with a thermal camera

The temperature of the micro-PINGUIN instrument is measured with a thermal camera and a Pt100 temperature probe as a reference. The thermal camera detects the infrared radiation emitted by an object, in this case, a microtiter plate well, and converts it into a visual image. As objects with a higher temperature emit more infrared radiation than objects with a lower temperature and as the temperature increased due to latent heat release once the droplet nucleates, this technique can be used to measure the freezing temperatures of the samples. The camera is sensitive within a wavelength range of 7.5 to 14.0 µm and has a resolution of 640×480 pixels. However, while the thermal camera has a high relative precision for the temperature reading (0.06 °C, Appendix A6), it has a low absolute precision of $\pm 2$ °C; therefore, a fix-point cavity is used as a reference measurement. The fix-point cavity is a copper tube with an angled bottom and a black inner surface. The radiation measured by the camera is scattered inside the cavity, which allows for a precise reading of the actual temperature $T_{cavity}$. The Pt100 reference temperature probe is positioned directly against the fix-point cavity and therefore measures the temperature $T_{Pt100}$ at the same position in the upper cooper base.

This temperature measurement is used as a reference temperature and its offset is applied to the temperature reading of the camera during the cooling experiment $T_{camera}$:

$$T_{corrected} = T_{camera} + (T_{Pt100} - T_{cavity}).$$

Further, the Pt100 temperature measurement serves as the input for the PID-regulated Peltier elements during the heating and cooling of the instrument. During the freezing of the sample, latent heat is released by the sample because of the phase change from water to ice. The initial phase of freezing when ice crystals start to form is a fast process resulting in an immediate

temperature increase in the sample to 0 °C. If this phase change is ongoing, the temperature stays at 0 °C and a plateau forms. When the sample is completely frozen, the release latent heat stops, and the droplet cools down to ambient temperature. This results in a characteristic temperature profile for a freezing event as shown in Fig. 4(b). The length of this plateau at 0 °C is among other factors dependent on the temperature where the nucleation is initiated. For nucleation events close to 0 °C, which is the case for some highly active biogenic INPs, the nucleation temperature and the temperature where the droplet is completely frozen can differ significantly (Supplementary S3). By using an infrared camera for the detection of the freezing event, we detect this immediate temperature increase upon nucleation as the freezing temperature of the sample. This is an advantage compared to the freezing point detection based on the change in the optical properties of the droplet. As long as the phase change is ongoing, the optical properties of the droplets can change, which makes it difficult to identify the exact moment of nucleation. Such variations in freezing point detection are especially crucial when investigating highly active INPs such as biogenic INPs.

Thus, in the micro-PINGUIN instrument, a thermal camera captures an image of the PCR plate every 5 seconds and after each run, the data is processed by a custom-made software. This is done as follows: (1) A grid is created by the user, making sure that the location of every well is marked in the program (Fig. 4(a)). (2) The temperature profile is then processed for each well and the freezing event is detected as a change in the slope of the temperature of each well (Fig. 4(b)).

As the change in temperature is smaller for freezing events close to 0 °C, these temperatures can cause problems in the automatic recognition of the nucleation temperature. By default, a freezing event is recognized when the temperature gradient shows a deviation in the temperature profile that is larger than two times the standard deviation. This value can be lowered to identify nucleation events at high temperatures. To avoid false detections, the value with the largest deviation in temperature is always used as the freezing temperature.

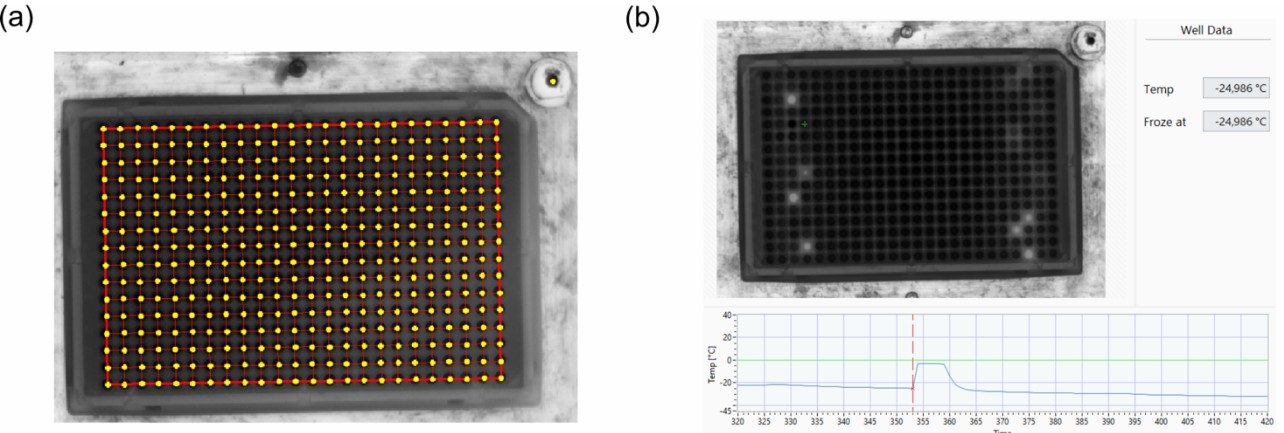

**Figure 4:** (a) Mask created for the freezing point detection. The yellow circles mark the pixels taken for the analysis. (b) Temperature profile of the droplet marked in green. The red line indicates the point in time with the highest temperature gradient.

## 2.4 Data analysis

Methods to detect droplet freezing events rely on dividing samples into multiple equal volumes and observing the freezing process of these volumes at varying temperatures (known as freezing curves), while maintaining a consistent cooling rate. However, it is important to note that droplet freezing assays have limitations in distinguishing solely between the liquid and frozen states of the droplets. Once the most active INP, among the mixture of INPs, initiates freezing and prompts droplet crystallization, the influence of less active INPs is hidden. To comprehensively investigate the freezing characteristics of highly

active samples across a broad spectrum of temperatures, it is necessary to study dilutions of the sample. Thereby, the most active INPs will be outdiluted, enabling the analysis of the INPs that are only active at lower temperatures. Calculating the quantity of INPs based on the proportion of frozen droplets, requires considering the particle concentration within the solution, the volume of the droplets, and the dilution factor. Following the approach by Vali (1971) and assuming time independence of freezing, the cumulative spectrum $K(T)$, corresponding to the number of sites active above temperature $T$ per unit sample

volume is described by the following equation;

$$K(T) = - \frac{1}{V} \ln \left( 1 - \frac{N_f(T)}{N_0} \right),$$

( 1 )

where $V$ is the droplet volume, $N_f(T)$ is the number of frozen droplets at a given temperature $T$ and $N_0$ is the number of total

240 droplets per sample. Based on this equation the number of ice nucleation active sites per mass of sample $n_m(T)$ is calculated as

$$n_m(T) = - \frac{1}{V} \ln \left( 1 - \frac{N_f(T)}{N_0} \right) \left( \frac{d}{c_m} \right),$$

using the dilution factor $d$ and the particle mass concentration $c_m$.

## 2.5 Measurement accuracy

The measurement accuracy of the micro-PINGUIN instrument is determined relative to a calibrated temperature standard. The individual components that contribute to the temperature uncertainty of the instrument are listed in Table 1 and were examined in separate experiments. A detailed description of the measurements and analysis is given in Appendix A1 to A6.

We found that the largest contribution to the uncertainty of the instrument is the vertical gradient within the well. As the freezing temperature of the sample is determined by the infrared camera measurement which is a surface-sensitive technique, any vertical gradient in the well will lead to an uncertainty in the temperature measurement. The total vertical gradient was 0.20 °C at 0 °C and increased by 0.015 °C per degree when the temperature was lowered. Given that temperature measurements are performed at the surface, all temperature readings should be corrected by half the vertical gradient at a given temperature, resulting in a symmetrical contribution. Thus, the freezing temperatures determined within the experiments are corrected by this temperature correction $T_{\text{correction}}$

$$T_{\text{correction}} = (0.021 \cdot \Delta T + 0.19)\ °C,$$

where $\Delta T$ is the difference between the room temperature $T_R$ (22 °C ± 1 °C) and the surface temperature $T_S$ measured by the infrared camera in the well $\Delta T = T_S - T_R$. The vertical gradient was not dependent on the cooling rate in the range between 0.3 °C min$^{-1}$ to 3 °C min$^{-1}$ (Supplementary S4) and thus, we conclude that the gradient is not attributed to a poor thermal conductivity between the individual parts of the instrument but rather to the warm air above the sample surface.

The individual uncertainty contributions result in an overall temperature-dependent standard uncertainty (k=1) for measurements with micro-PINGUIN of

$$\delta T = \sqrt{(\delta T_V)^2 + (\delta T_T)^2 + (\delta T_{TC})^2 + (\delta T_{TL})^2 + (\delta T_{TR})^2 + (\delta T_{CD})^2 + (\delta T_{CR})^2 + (\delta T_{NUC})^2}$$
$$= \sqrt{(0.012 \cdot \Delta T)^2 + 0.017}\ °C.$$

Exemplary uncertainty and correction values for different temperatures are given in the Table A1. The horizontal gradient of the instrument was below the sensitivity of the infrared camera (< 0.06 °C) and therefore not included in the calculations. Further factors such as the deviation of the fix-point cavity from a black body, or the thermal anchoring between the fix-point cavity and the Pt100 temperature probe are considered to have only minor impact on the accuracy of the instrument and were therefore not investigated in detail here.

**Table 1:** Quantities contributing to the temperature uncertainty of micro-PINGUIN. The standard uncertainty was determined experimentally and the contribution to the uncertainty is estimated by taking the distribution of the uncertainty into account.

| Quantity | Description | Uncertainty contribution |
|---|---|---|
| $\delta T_V$ | vertical gradient | $\sqrt{(0.012 \cdot \Delta T)^2 + (0.11)^2}$ °C |
| $\delta T_T$ | Pt100 uncertainty | 0.0081 °C |
| $\delta T_{TC}$ | Pt100 calibrator uncertainty | 0.0080 °C |
| $\delta T_{TL}$ | Pt100 long-term drift | 0.029 °C |
| $\delta T_{TR}$ | Pt100 repeatability | 0.0016 °C |
| $\delta T_{CD}$ | thermal camera distortion | 0.021 °C |
| $\delta T_{CR}$ | thermal camera repeatability | 0.052 °C |
| $\delta T_{NUC}$ | non-uniformity correction | 0.043 °C |

## 3 Ice nucleation activity of Snomax and Illite

### 3.1 Snomax

The characterisation of the micro-PINGUIN instrument involved the use of extensively researched materials, and the obtained outcomes were juxtaposed with outcomes from established ice nucleation instruments. As part of the INUIT (Ice Nuclei research UnIT) initiative, comparative assessments of various ice nucleation instruments were conducted, employing Snomax (Wex et al., 2015) and Illite NX (Hiranuma et al., 2015). Snomax is commercially available and consists of freeze-dried cell material of the ice nucleation active bacterium *Pseudomonas syringae* and is ice nucleation active already at temperatures as high as -2 °C (Wex et al., 2015). As the activity of Snomax was proposed to decrease over time, a new batch was ordered from the manufacturer and stored at -20 °C until usage. Care was taken that the Snomax batch was not subjected to many temperature changes, as recommended by the manufacturer. To cover the temperature range of INPs that are active at lower temperatures, Illite NX powder was used. Illite NX powder consists of different minerals including Illite, Kaolinite, Quartz, Carbonate and Feldspar. As the same batch of Illite NX is used as in the INUIT intercomparison study our results can be directly compared to the results reported by Hiranuma et al. (2015). Measurements with Snomax suspensions of concentrations ranging from $10^{-2}$ mg ml$^{-1}$ to $10^{-7}$ mg ml$^{-1}$ were repeated three times. Results are showcased in Fig. 5(a) alongside the data acquired by Wex et al. (2015). The measurements obtained with the micro-PINGUIN instrument show freezing of Snomax as high as -3.5 °C. At -12 °C, concentrations of INPs reach a plateau at $10^9$ INPs per mg of Snomax. At the knee point around -10 °C, INP concentrations obtained in this study are slightly below the values of previous measurements with other instruments, but the plateau reached by the curve is in agreement with that reported by Wex et al. (2015). We noted significant discrepancies in repeated Snomax measurements, even when employing the identical instrument and the same batch of Snomax. As Snomax

contains freeze-dried cells of *P.syringae* bacteria but also fragments of the cell membrane, remains of the culture medium and other unknown material, the number of INPs can vary within the prepared suspension leading to large variations in the measured freezing curves. Consequently, minor disparities in freezing spectra measured by diverse instruments are expected if different suspensions are used. Furthermore, the variance in ice nucleation activity could potentially stem from the utilization

of distinct batches of Snomax in the two studies, variations within the substrate or the possibility of marginal reduction in Snomax activity due to storage. We could significantly improve the reproducibility of the measurements by using aliquots of a Snomax suspension that were stored frozen until usage as shown in Fig. 5(b). This points at the key role that substance heterogeneity plays for measurement reproducibility. The lower onset freezing temperature in Fig. 5(b) is attributed to the large variability between freshly prepared Snomax suspensions and not due to a decrease in activity upon freezing. We

evaluated the impact of freezing and thawing to the suspension and found that the variations are within the measurement uncertainty (Supplementary S5 and S6). Thus, we propose the use of Snomax suspensions that are prepared in advance and stored frozen in aliquots for reproducibility measurements and further instrument intercomparison studies. The reproducibility of measurements using frozen aliquots is further discussed in Sect. 3.3.

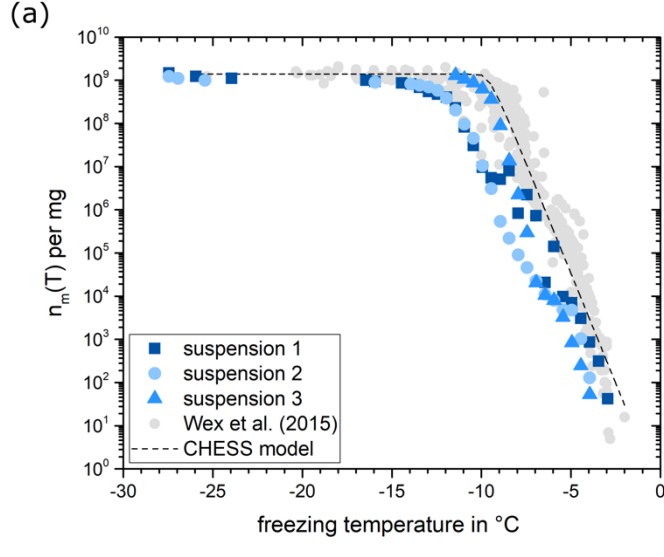

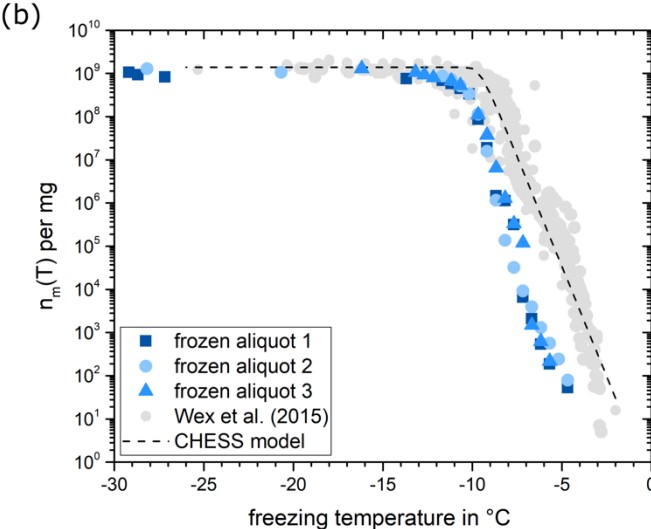

**Figure 5:** (a) Number of INPs per mg Snomax measured for suspensions with $10^{-2}$ mg ml$^{-1}$ to $10^{-7}$ mg ml$^{-1}$ Snomax prepared freshly on three different days. The data was binned in 0.5 °C temperature bins. Grey data points represent the results from various ice nucleation instruments investigated by Wex et al. (2015). The dashed line is based on the CHESS model by Hartmann et al. (2013). (b) The same is shown as in (a) but with the Snomax suspension that was prepared once, stored frozen in aliquots, and the measurement was repeated on freshly thawed
aliquots three times.

## 3.2 Illite NX

The number of INPs normalized to the surface area $n_{s,BET}(T)$ measured for Illite NX suspensions of concentrations between 10 mg ml$^{-1}$ and 0.1 mg ml$^{-1}$ is shown in Fig. 6 in comparison to data from other devices analysing Illite NX suspensions. $n_{s,BET}(T)$ was derived from the $n_m(T)$ spectrum following the approach by Hiranuma et al. (2015):

$$n_{s,BET}(T) = \frac{n_m(T)}{\theta},$$

with the specific surface area $\theta$ obtained from gas-adsorption measurements (BET-derived surface area) of 124.4 m$^2$ g$^{-1}$. We assume the same surface area, as we used an aliquot sourced from the same batch of Illite NX as the one used by Hiranuma et al. (2015). The data displayed in Fig. 6 presents three measurements with freshly prepared suspensions. The measurements conducted on Illite NX demonstrated fair reproducibility. On average freezing started at around -7.1 °C with a concentration of 5.1·10$^{-1}$ INPs per m$^2$ of Illite NX. The measurements obtained in this investigation fall in the lower end of the spectra recorded by other devices using a polydisperse Illite NX suspension (Hiranuma et al., 2015; Beall et al., 2017; Harrison et al., 2018; David et al., 2019). Thus, our data extends the concentrations of INPs reported for Illite NX suspensions.

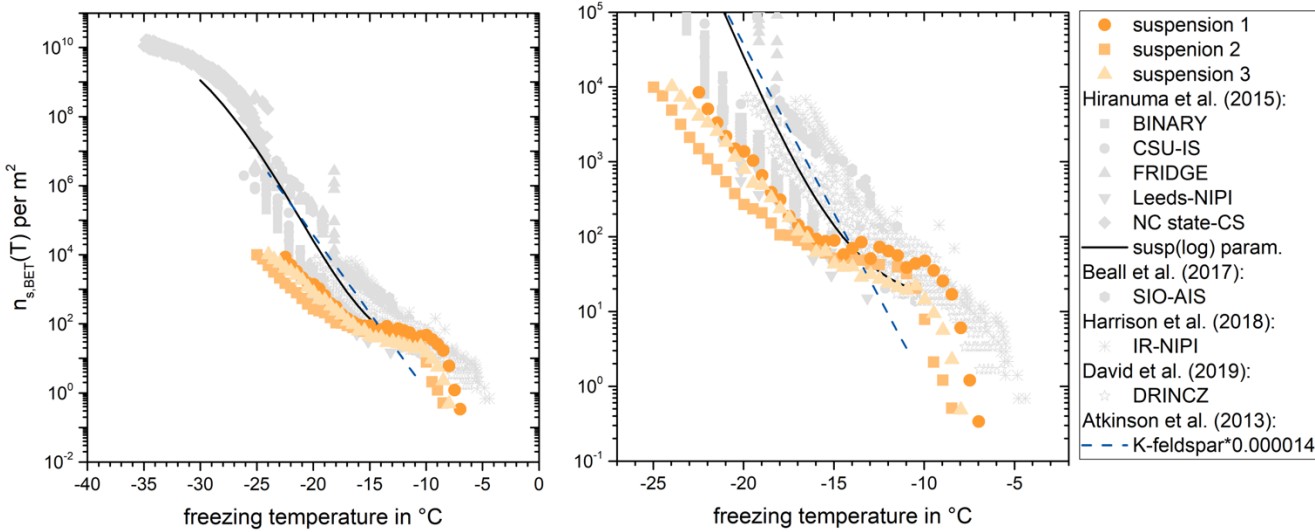

**Figure 6:** Left: Number of INPs per surface area of Illite NX (orange data). The grey data points represent measurements for comparable instruments (Hiranuma et al., 2015; Harrison et al., 2018; Beall et al., 2017; David et al., 2019). The black line shows the fit for all suspension measurement techniques in the logarithmic representation and the dashed blue line the fit from Atkinson et al. (2013) for K-feldspar multiplied with a factor of 0.000014 as discussed in Hiranuma et al. (2015). Right: The right panel shows the same data zooming in on the range 10$^{-1}$ m$^{-2}$ < $n_{s,BET}$ < 10$^5$ m$^{-2}$.

### 3.3 Reproducibility of the measurements

To assess the consistency of measurements conducted with the micro-PINGUIN instrument, successive experiments were carried out employing the identical suspension. This approach aimed to mitigate the influence of dilution errors and variations within the substrates that we used in the tests. Initial trials involving Snomax, Illite, and Feldspar suspensions revealed an aging phenomenon over the course of the day, despite storing the suspensions in the refrigerator between the individual experiments. As a result, the suspensions were freshly prepared immediately prior to each experiment, or the suspension was divided into aliquots that were frozen for preservation until needed. In the latter case, the samples were thawed just before conducting the ice nucleation experiment, and all measurements were executed within the same day to minimize disparities in freezer storage time. For characterizing this procedure, Snomax was chosen as the test substance due to its biogenic origin and the notable deviations in previous measurements. Figure 7 shows the mean value and standard deviation of the fraction frozen curves for three measurements with a concentration between $10^{-2}$ mg ml$^{-1}$ and $10^{-7}$ mg ml$^{-1}$ Snomax. The experiment's standard deviations range from 0.006 °C to 1.191 °C, showing outliers for exceedingly high and low INP counts, respectively. Bacterial ice nucleating proteins show distinct freezing behaviour depending on the size of the proteins and can be divided into different classes (Turner et al., 1990; Yankofsky et al., 1981; Hartmann et al., 2013; Budke and Koop, 2015). Budke and Koop (2015) identified two classes of INPs for Snomax, the high active but less abundant class A INPs nucleate ice at around -3.5 °C, while class C INPs are frequently observed but nucleate at lower temperature of -8.5 °C. Reproducibility within freezing temperatures ranging from -7 °C to -10 °C outperformed that at higher temperatures, likely due to the prevalence of class C INPs over the class A, which results in a higher class C homogeneity. The mean standard deviation for this dataset is 0.20 °C, leading to an estimated reproducibility of ± 0.20 °C for the micro-PINGUIN instrument. We suggest that this is a conservative estimate due to the difference in the prevalence of class A and class C INPs in the sample. Thus, while the standard deviations observed for the temperature range, where class A INP show predominant activity, are due to a combined effect of the technical reproducibility of our instrument and inhomogeneity of the INPs across the droplets, the standard deviations observed for the temperature range, where class C INP show predominant activity, reflect primarily the technical reproducibility of our instrument. The results of these reproducibility measurements are in agreement with the measurement uncertainty determined earlier. Further, we could demonstrate that the reproducibility of the measurements is greatly improved when suspensions are stored frozen in aliquots. Storage of the sample for 4 months at -20 °C resulted in a slightly higher standard deviation for the freezing curves but no clear reduction in ice nucleation activity was observed. The freezing spectra were partly within the standard deviation of the three initial measurements while other dilutions showed slightly higher or lower freezing temperatures (Supplementary S7 and S8). Overall, the reproducibility was improved compared to freshly prepared Snomax suspensions.

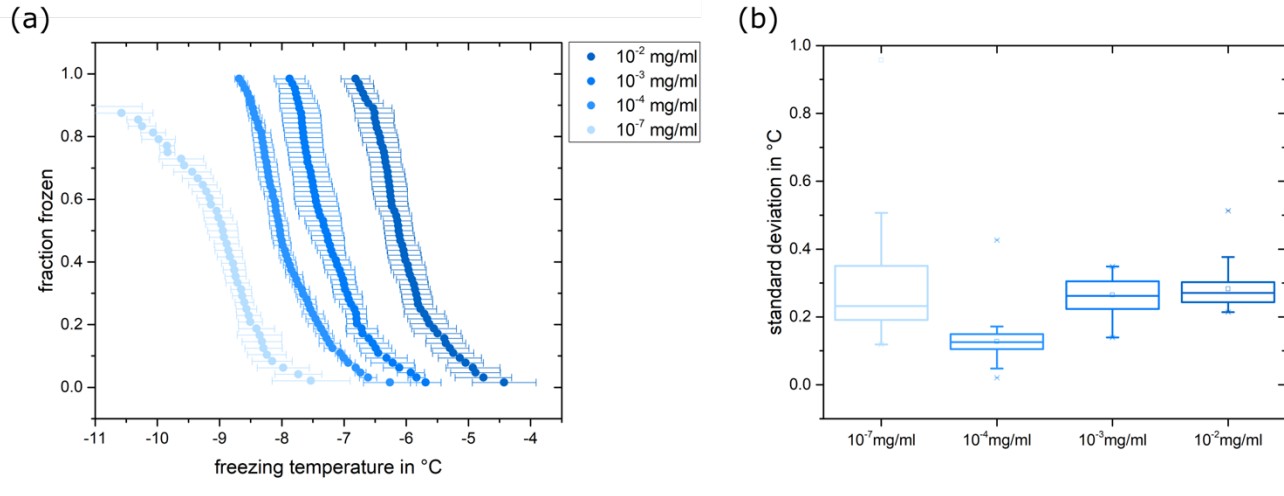

**Figure 7:** (a) Fraction frozen for Snomax suspensions with a concentration between $10^{-2}$ mg ml$^{-1}$ and $10^{-7}$ mg ml$^{-1}$. The data for $10^{-5}$ mg ml$^{-1}$ and $10^{-6}$ mg ml$^{-1}$ are not shown for illustrative purposes. Data points represent the mean values of three measurements and the horizontal error bars indicate the standard deviation between these three measurements. (b) Standard deviations for the different concentrations are shown as box plots with 25th and 75th percentiles.

## 4    Conclusion

We developed a novel ice nucleation instrument that supports accurate nucleation temperature detection within the temperature range of 0 °C to -25 °C. The distinctive feature of this instrument is the utilization of a gallium bath, which acts as a platform that holds the PCR plates with the samples. The gallium bath ensures tight contact between the sample and the surrounding cooling unit and thus, resulting in good thermal conductivity. Further, the freezing events are detected with high precision by an infrared camera based on a sudden rise in temperature following the nucleation event. This facilitates the recognition of nucleation events instead of freezing events, further reducing the uncertainty in assigned freezing temperatures. The instrument was thoroughly analyzed for its reproducibility and accuracy of the temperature measurements and therefore can be used for reliable and intercomparable ice nucleation studies. Based on the reproducibility experiments, we recommend that the Snomax suspensions are prepared in advance and stored frozen in aliquots for future reproducibility and instrument intercomparison measurements.

## Appendix A: Description of the uncertainty measurements

### A1. Vertical temperature gradient $\delta T_V$

The vertical temperature profile measurements were performed with a thin thermistor (PSB-S9 Thermistor, PB9-43-SD6) to minimize the disturbance of the measurement by the thermistor and to allow for measurements at several depths in the well. The wells of the 384-well PCR plate were filled with 30 µl of sterile filtered MilliQ water and the thermistor was mounted on a micromanipulator positioned above the well. The first measurement was performed with the thermistor approximately 1 mm below the water surface. The cooling experiment was started with 1 °C min$^{-1}$ until -15 °C, while the temperatures measured with the small thermistor inside the well and a reference temperature probe are recorded by the instrument. After the cooling cycle, the instrument reached a steady state temperature at around 2 °C (temperature of the cooling water), and the thermistor was lowered by 1.5 mm using the micromanipulator.

The gradient measurements were performed both in the center and at a corner of the 384-well PCR plate. The temperature profiles were recorded for several depths and then evaluated relative to the reference temperature probe. As some freezing events occurred during the measurements, a linear regression analysis was performed for the temperature profiles in the temperature range between 0 °C to -6 °C to determine the vertical gradient of the instrument.

We observed slightly more gentle gradients at the corner of the plate than in the center of the plate, and thus the steepest gradient was used to estimate the total vertical gradient of the instrument. The vertical gradient contribution in the center well was measured to 0.20 °C at 0 °C and increased 0.015 °C per degree with lower temperatures (Fig. A1). Given that temperature measurements are done at the surface, all temperature readings should be corrected by half the vertical gradient at a given temperature, resulting in a symmetrical contribution. During the measurements, we observed that the temperature reading is slightly different if the thermistor is touching the wall of the well, probably due to the different thermal conductivity of the plastic material. Due to the conical shape of the wells, it was not possible to lower the thermistor further without contact to plastic wall. Due to these limitations, the gradient measurement covers only a depth of approximately 3 mm and would be ideally extrapolated to cover the full depth of the well. To be conservative in our evaluation, we assume a linear relationship and therefore multiply the vertical gradient by a factor of 3/8 as the sample in the well has a depth of approximately 8 mm. Thus, the temperature correction due to the vertical gradient is given by the following equation:

$$T_{\text{correction}} = \Delta T \cdot 0.021 + 0.19 \text{ °C},$$

where $\Delta T$ is the difference between the room temperature $T_R$ (22 °C ± 1 °C) and the surface temperature $T_S$ measured by the infrared camera in the well $\Delta T = T_S - T_R$. The contribution of the vertical gradient represented as the standard uncertainty can be expressed by the following equation:

$$\delta T_V = \sqrt{(0.012 \cdot \Delta T)^2 + (0.11)^2} \text{ °C}.$$

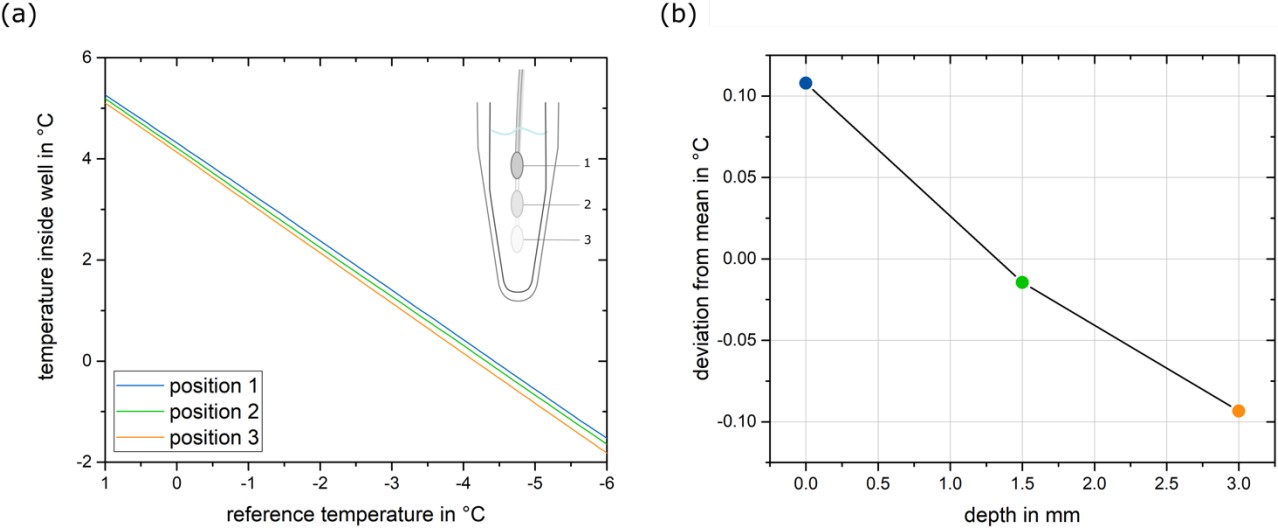

**Figure A1:** Vertical gradient measurements for well G13 in the middle of the 384-well PCR plate. (a) temperature profiles measured for 3 positions in the well. (b) deviation of the temperature from the mean temperature reading at 0 °C for the three depths.

## A2. Pt100 uncertainty $\delta T_{\mathrm{T}}$, long-term drift $\delta T_{\mathrm{TL}}$ and Pt100 calibrator uncertainty $\delta T_{\mathrm{TC}}$

420 The temperature measurement of the Pt100 temperature probe is based on a change in its resistance as a function of temperature change and is measured using a National Instruments measurement module (NI-9219, National Instruments, US). Calibration correction parameters were determined using an AMETEK reference temperature calibrator (RTC-157, AMETEK, US) with an external reference temperature probe (Pt100 Resistance Probe, 652747-11) within the temperature range of 35 °C to -35 °C. The micro-PINGUIN Pt100 temperature probe was submerged in ethanol inside the RTC calibrator and data was recorded for

425 5 min at steady-state conditions in 5 °C intervals. This results in a maximum residual mean error of 0.0081 °C within the temperature range of 35 °C to -35 °C. Additionally, the uncertainty of the calibration device $\delta T_{\mathrm{TC}}$ must be considered, which is given by a calibration certificate ($\delta T_{\mathrm{TC}} = 0.02$ °C). After a 2-month operation time, the temperature probe will be recalibrated to determine the long-term drift of the temperature reading $\delta T_{\mathrm{TL}}$. The manufacturer states a long-term stability of better than 0.05 °C per 5 years. Initially, the micro-PINGUIN instrument was equipped with a thermistor for the reference temperature

430 measurement. However, during the detailed examination of the uncertainty we noticed that the thermistor had a high long-term drift. Thus, the thermistor was replaced by the aforementioned Pt100 temperature probe. The vertical gradient measurements and the examination of the infrared camera repeatability, distortion and non-uniformity correction were conducted with the previous thermistor probe, however the results are not influenced by the exchange of the reference temperature probe. The measurements by both sensors are accurate and the exchange was only due to the better long-term

435 stability of the Pt100 probe.

### A3. Pt100 repeatability $\delta T_{TR}$

After the calibration, the stability of the Pt100 temperature probe reading was examined by recording the temperature at steady state conditions at 0 °C for 3 minutes. The standard deviation of the temperature reading was calculated to 0.0016°C and is used to derive the temperature probe repeatability contribution $\delta T_{TR}$.

### A4. Thermal camera repeatability $\delta T_{CR}$

For accurate results, the camera's manufacturer recommends powering up the camera before the measurement. We evaluated the camera warm-up time by recording the deviation between the temperature measured by the infrared camera relative to the reference temperature probe at the fix-point cavity (calibration offset). This experiment was performed at steady state conditions (room temperature) for one and a half hours after powering the camera. Improved stability of the temperature measurement was found after a 40 minute operating time of the camera. The repeatability of the temperature reading by the infrared camera is calculated by the standard deviation of the temperature reading at steady-state conditions recorded after the warm-up period. The standard deviation of the temperature readings is ± 0.05 °C.

### A5. Non-uniformity correction $\delta T_{NUC}$

The infrared camera is constantly recording its internal temperature and performing a non-uniformity correction to account for minor detector drifts that occur over time due to internal temperature changes. We observed that the temperature measured before and after this correction can differ slightly. Thus, we evaluated this impact by manually performing several non-uniformity corrections while recording the temperature at steady-state conditions (room temperature). The non-uniformity correction contribution was measured to 0.15 °C.

### A6. Thermal camera distortion $\delta T_{CD}$

The inhomogeneity of the camera lens has an impact on the temperature measurements across the PCR plate. To evaluate this contribution, the camera was attached to a movable plate and the black body radiation of the fix-point cavity was measured for several positions of the camera at steady-state conditions. The offset between the infrared camera and the temperature measured by the temperature probe for several positions gives an estimate of the lens distortion of the camera. The average temperature offset for each position is calculated over a 2 minute measurement period with an image frequency of 120 images per minute to minimize the impact of repeatability contributions. The lens distortion contribution was measured to 0.06 °C.

## A7. Measurement uncertainty values

**Table A1:** Measurement uncertainty and temperature corrections for various temperatures in 5 °C steps. The measurement uncertainties are expanded with to a coverage of 95%.

| Temperature | Correction | Uncertainty (k=2) |
|---|---|---|
| 0 °C | -0.27 °C | ± 0.59 °C |
| -5 °C | -0.38 °C | ± 0.70 °C |
| -10 °C | -0.48 °C | ± 0.81 °C |
| -15 °C | -0.59 °C | ± 0.93 °C |
| -20 °C | -0.69 °C | ± 1.04 °C |
| -25 °C | -0.80 °C | ± 1.16 °C |


*Author contributions.* All authors contributed to the development and design of micro-PINGUIN. The instrument was manufactured and tested by CM and MRJ. The ice nucleation and uncertainty experiments were conducted by CW. MRJ developed the analysis software and calculated the uncertainty budget. The paper was written by CW, with contributions from all co-authors. The project was supervised by TST and KF.

*Competing interests.* The authors declare that they have no conflict of interest.

*Acknowledgements.* We are very grateful to Andrey Chuhutin, Frederik Voldbirk, Peter Melvad, Lorenz Meire and Sigurd Agerskov Madsen for their contributions to the development of the initial instrument. The authors would like to thank Jeppe Fogh Rasmussen for providing the small thermistors and his assistance with the gradient measurements. Further, we acknowledge Mikkel Bo Nielsen for his guidance on the temperature and accuracy measurements. We thank Heike Wex for providing data of the Snomax intercomparison study and Alexei Kiselev for providing data and the Illite samples.

*Financial support.* This work was supported by The Villum Foundation (23175 and 37435), The Carlsberg Foundation (CF21-0630), and The Novo Nordisk Foundation (NNF19OC0056963).

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
