# Peer review of "Micro-PINGUIN: Microtiter plate-based ice nucleation detection in gallium with an infrared camera"

_EGUsphere, 2024_

## Author Comment (AC1)

**Referee comments #1**

**Major comments:**

1. The authors recommend storing Snomax suspensions in aliquots. While I agree that the ice nucleation activity (INA) of Snomax suspensions with such a treatment shows improved reproducibility among repeating experiments. It's noted that its INA has a lower value compared to the freshly prepared samples (Figure 5). This indicates that the sample experiences changes during storage. Therefore, the statement that aliquot storage is a better treatment for preserving Snomax samples requires further justification.

   Thank you for this comment. It is correct that the INA activity of the frozen Snomax is starting at lower temperatures compared to the curve shown in Figure 5(a). However, we attribute this to the variability between the suspensions. We have done an additional experiment to evaluate the impact of freezing and thawing on the Snomax suspension and found that the activity is not reduced by freezing. We have added Figure S5 to the Supplementary and the following sentence in the manuscript:

   "The lower onset freezing temperature in Fig. 5b is attributed to the large variability between freshly prepared Snomax suspensions and not due to a decrease in activity upon freezing. We evaluated the impact of freezing and thawing to the suspension and found that the variations are within the measurement uncertainty (Supplementary S5)."

[Figure]

2. The authors claim that the INAs of illite NX obtained in the present studies are comparable to those reported by previous studies. However, a difference in INAs obtained between the two datasets can reach up to 3 orders of magnitude (Figure 6). This conclusion on "The micro-PINGUIN instrument was validated using the well-studied substances Snomax and Illite NX and the results obtained in this study are consistent with already existing instruments." needs justification.

   We agree that the measurements presented in this study are up to 3 orders of magnitude lower than the previous studies shown as a grey background. However, also

between these previous studies we find that the measurements differ by up to 3 orders of magnitude. Within the temperature range from -5°C to -10°C, so far only one instrument showed activity (Harrison et al., 2018), therefore the spread for higher temperatures is not yet defined. Thus, we argue that the results are consistent with what was found in previous studies. Further, we have corrected the temperatures by half of the vertical gradient to account for the temperature deviation within the well. This was not done by studies using other instruments that employ larger volumes, which would have shifted the presented freezing curves to lower temperatures (Table A1).

3. The temperature bias introduced by the inherent uncertainties from Pt 100 is not indicated in the present work.

In the Appendix A2 and A3 we discuss the measurement uncertainties of the Pt100 with regard to the long-term drift, the uncertainty and repeatability of the PT100 and the uncertainty of the calibration device. We have rewritten the section A3 to clarify the Pt100 repeatability measurements.

"After the calibration, the stability of the Pt100 temperature probe reading was examined by recording the temperature at steady state conditions at 0°C for 3 minutes. The standard deviation of the temperature reading was calculated as 0.0016 °C and is used to derive the temperature probe repeatability contribution $\delta T_{TR}$."

4. Specific details are absent in the schematic of the experimental setup, as outlined below.

**Minor comments:**

1. Comments on figures: (1) For all figures: The name of each component can be indicated in the figure for clarity reasons.  (2) Figure 1: Components E and D are not connected to any other units, can you make it more specific? If B is "water cooling", how come the temperature of B goes to -2 °C?

We changed the figures accordingly as shown below.

[Figure]

(3) Figure 2: What are the red and blue bars, current or circulating cooling fluid?

We updated the figure legend to clarify that the bars are showing the circulation of cooling liquid. *"**Figure 1:** Schematic drawing of the cooling base. The red and blue tubes connected to the water cooler base indicate the circulation of cooled water which removes the heat generated by the Peltier elements."*

[Figure]

(4) Figure 3: Can you also specify the flow rate in the legend?

We have updated the figure legend.

[Figure]

(5) Figure 7: clarify the meaning of horizontal error bars – whether they represent standard deviations or the previously mentioned temperature uncertainties.

We have updated the figure legend *"**Figure 2:** (a) Fraction frozen for Snomax suspensions with a concentration between $10^{-2}$ mg ml$^{-1}$ and $10^{-7}$ mg ml$^{-1}$. The data for $10^{-5}$ mg ml$^{-1}$ and $10^{-6}$ mg ml$^{-1}$ are not shown for illustrative purposes. Data points represent*

*the mean values of three measurements and the horizontal error bars indicate the standard deviation between these three measurements. (b) Standard deviations for the different concentrations are shown as box plots with 25th and 75th percentiles."*

2. L170: What do you mean by "dry air"? What are the components of the dry air and how was it produced? Why the largely deviated freezing curve was not observed in other studies, most of which are used under room temperature and relative humidity conditions? Do you have any films to isolate the suspensions and air?

   The dry air is air with a humidity <10%RH that is provided from a centralized compressor suppling the entire building. Due to the use of an infrared camera, we cannot cover the droplets as done in same other studies. Previous studies describe a similar approach: for example, Schiebel et al. (2017) and Budke and Koop (2015) who apply a flow of dry air or $N_2$ to the instrument to avoid frost formation.

   We modified the sentence in line 161: "Tests with a flow of compressed air with a low humidity (<10 % RH) passing through the camera tower showed that condensation was avoided during the experiment and that the freezing temperatures decreased with increasing air flow until $T_{50}$ temperatures, corresponding to the temperature where 50 % of the droplets are frozen, of around -25 °C were reached (Fig. 3)."

3. L259: Do you mean the thermal conductivity between gallium and PCR trays?

   We have modified the sentence to clarify this point "The vertical gradient was not dependent on the cooling rate in the range between 0.3 °C min$^{-1}$ to 3 °C min$^{-1}$ (Supplementary S3) and thus, we conclude that the gradient is not attributed to a poor thermal conductivity between the individual parts of the instrument but rather to the warm air above the sample surface. "

4. L125: Ensure consistency in referring to the "temperature probe" by specifying if it corresponds to the "pt100 probe" throughout the main text.

   We have exchanged "temperature probe" by "Pt100 temperature probe" throughout the text.

5. Figure 3: Will higher flow rates>20 L/min introduce any changes in the freezing curve? I wonder if 20 L/min represents the optimal condition for your measurement.

   In section 2.2 we describe the airflow system. The 20 L/min flow shown in Figure 3 is only used to evaluate the impact of a dry air flow. The conditions during the experiment are described in line 170: "Before each run, the camera tower is flushed with a high flow of dry air (20 l min$^{-1}$). The flow is reduced to 10 l min$^{-1}$ during the measurement to minimize disturbance of the samples and the introduction of warm air."

   We added the following sentence to clarify this point: "We measured the relative humidity in the camera tower for this procedure and found a flow of 10 l min$^{-1}$ is sufficient to maintain a low relative humidity."

6. L352: Define "type A and type C INPs" for better comprehension. Provide an explanation for these terms.

We added the following explanation in line 351:
"Bacterial ice nucleation proteins show distinct freezing behaviour depending on the size of the proteins and can be divided into different classes (Turner et al., 1990; Yankofsky et al., 1981; Hartmann et al., 2013; Budke and Koop, 2015). Budke and Koop (2015) identified two classes of INPs for Snomax, the high active but less abundant class A INPs nucleate ice at around -3.5 °C while class C INPs are frequently observed but nucleate at lower temperature of -8.5 °C."

---

## Author Comment (AC2)

**Referee #2**

1. Perhaps birch pollen washing water, freezing within a particularly narrow temperature window from -17 °C to -18 °C (Häusler et al., 2018, https://www.mdpi.com/2073-4433/9/4/140), would have been a good third substance to test.

   Thank you for this suggestion. It would be interesting to compare the freezing of birch pollen washing water, however this is beyond of the scope of this study. We have decided to use Snomax and Illite as test substance due to the already existing intercomparison studies.

2. A total of 384 droplets provides the opportunity to derive differential freezing spectra (Vali, 2019, https://doi.org/10.5194/amt-12-1219-2019) that eventually show the different types of INP discussed in Section 3.3. With little effort a re-analysis of the available data may thereby yield additional insights into ice nucleation active components of snomax and illite that can hardly be gleaned from the cumulative spectra, e.g., in Figures 5, 6, 7, S4, and S5..

   Thank you for this suggestion. In our experiments, the 384-well plates were divided to measure several dilutions in the same run. In this way, we measured between 48 to 80 droplets per sample. As the main aim of the article was to validate the PINGUIN instrument, we chose to present cumulative spectra which are also used in intercomparison studies that we refer to.

3. The infrared camera is said to detect the moment of an 'ice nucleation event' (line 68), whereas an optical camera observes a prolonged period, that is the 'change in optical properties such as brightness of the sample during the process of the whole droplet freezing' (lines 68 and 69). Right, but the 'ice nucleation event' can nevertheless be located in time at the beginning of changes in optical properties, no matter how long it took until the droplet was completely frozen. Perhaps I am wrong here, but I would expect to see in the infrared camera record at a warm freezing temperature, say at -4 °C, a rise in droplet temperature that is not a sudden step change and that also leaves some room for interpretation regarding the exact onset of freezing. The temperature record shown in Figure 4 is hard to analyse in this respect. Could you please show instead the record of a droplet frozen near -4 °C, and narrow the range of the time axis to the minute or so in which the peak occurred?

   During the nucleation event the droplet temperature rises to 0°C within around 10 seconds. This temperature increase is fast also at high freezing temperatures. However, at high freezing, it takes longer for the droplet to freeze completely and cool down to ambient temperature. Although, it takes around 10 seconds to reach the plateau at 0°C, we can detect the nucleation event very precise at the starting point of this temperature increase. We have added Figure S3 to the Supplementary showing the temperature profile for various freezing temperatures. As the temperature increase is higher for droplets freezing at lower temperature, we decided to show a droplet freezing at -25°C in Figure 4. However, we have changed the x-axis to show the freezing event in more detail.

[Figure]

4. Lines 36 and 55: Replace 'high fraction' with 'large fraction' and 'high number' with 'large number'.

   We have changed the sentences as suggested.

5. Line 56: Replace 'low number' with 'small number'.

   We have changed the sentence as suggested.

6. Lines 90 to 92: Consider rearranging the sentence in this way: 'Furthermore, we address the challenges due to inhomogeneities of the product and due to aging effects and propose a possible solution for using Snomax as a suspension for intercomparison studies and reproducibility measurements.'

   We have changed the sentence as suggested.

7. Figure 2: Better use a colour for the vapour chamber (G) that is different from that of the copper components above and below it.

We have updated the figure and changed the color of the vapor chamber.

[Figure]

8. Line 150 onwards: I appreciate the idea to heat the samples for repeated analysis, but how can evaporative loss be prevented, especially in heat treatments near boiling point?

   Thank you for this thought. We would need to evaluate this factor once the system modifications allow that heat treatments. If evaporation is a problem, we would cover the PCR plates with an adhesive plastic foil during the heating process.

9. The same question about evaporation arises in the next section, where the flow of dry air is discussed. During the development of the procedure, were sample trays weighed before and after a 40 min run to assess the loss due to evaporation?

   Thank you for this suggestion. We have now performed the suggested experiment and found that the loss due to evaporation is 0.36% of the volume and thus negligible. We have added the following sentence after line 180:

   "We have evaluated the sample loss due to evaporation and found that this factor is negligible as only 0.36% of liquid was lost during an experiment."

10. Line 175: A 'was' is missing before 'usually'.

    Thank you, we have added the missing word.

11. I am not sure whether Section 2.4 is needed because it mostly describes common practice. Consider reducing it to the bare minimum and merging it with the preceding section.

    Thank you for this suggestion. We agree that it is common practice, however, we have decided to include a short description to the manuscript to motivate and introduce the equations.

12. Figure 5a: At T > -11 °C error bars extend to 1 INP/mg snomax, suggesting that one of the three experiments no or very little freezing events were observed > -11 °C. After looking at Figure S4, I understand this is an artefact caused by the assumption of a normal distribution. In principle, you could estimate the multiplicative standard deviation (Limpert et al., 2001, https://academic.oup.com/bioscience/article/51/5/341/243981). However, three replicates cannot provide an estimate for that. Therefore, better show in Figure 5 all three replicates and not their mean and normal standard deviation.

We have updated Figure 5 and Figure 6 accordingly and show the 3 individual experiments instead of the mean and standard deviation.

[Figure]

Figure 6:

[Figure]

13. Line 321: Consider to replace 'using' with 'analysing'.

We have changed the sentence accordingly.

14. Line 330: The statement 'are within the range of the concentrations reported therein' has to be narrowed to the temperature range in which it actually applies (-8 °C to -23 °C).

We have modified the sentence: "The measurements obtained in this investigation fall in the lower end of the spectra recorded by other devices using a polydisperse Illite NX suspension (Hiranuma et al., 2015; Beall et al., 2017; Harrison et al., 2018), but are within the range of the concentrations reported therein for temperatures from -8 °C to -23 °C"

15. Figure S5: For fresh and old samples use better distinguishable symbols, e.g. , open circles and crosses, respectively.

We have updated the figure.

[Figure]

16. Lines 376 and 377: Maybe reconsider the statement 'recognition of nucleation events instead of freezing events' (please see my earlier comment above on this issue).

Due to the above explanation, we argue that this statement is true.

17. Line 380: Not sure what is meant by 'intercomparable' here, perhaps 'comparable'?

With "intercomparable" we want to express that it is comparable between the studies.

---

## Author Response (AR2)

Referee 1

1. The authors attributed the lower INA activity of the frozen Snomax compared to the fresh sample to the variability between the suspensions. Given that they have conducted the additional experiments, I would suggest they add the uncertainty of the calculated nm based on the repeating experiments. Also, it's good to include the uncertainties reported by Wex et al. [2015]. This would help confirm that the lower values are indeed due to the variability between the suspensions.

Thank you very much for this suggestion. We were advised by referee 2 to remove the standard deviations from the graphs, thus we decided to show the data for the three individual experiments instead. We have now again added the graph that shows the average and standard deviations of the experiments to the Supplementary S5. In Figure S5(b) the standard deviation for 3 freshly prepared suspensions is shown. The activity of the frozen suspension falls within this range when considering the large error bars. We appreciate the suggestion to include the uncertainties by Wex et al., however the data is not publicly available. Further, the data from the intercomparison study shows the deviation between various instruments and additionally the variation within the substrate. Thus, the uncertainty would consequently be combined for these two effects and therefore not comparable with what we show in this study.

[Figure]

2. While I agree that the active site surface density (ns) reported by previous studies varied within three orders of magnitude, upon considering the uncertainty of ns measured in their studies and the variation of ns reported by previous studies, the measured ns still falls within the lower limit of the ns range reported by previous studies, particularly at colder temperatures. I will not say they are comparable, at least based on the showed figure 6. In addition, David et al. [2019] have also reported the ns of illite NX at temperatures from -10 °C to -5 °C. At lower temperatures, Chen et al. [2018] have also reported ns illite NX at lower temperature ranges. More studies can be included for comparison.

We have reformulated the sections in the manuscript that state comparability to the previous data. Further, we have added the ns data of Illite from David et al. (2019) in Figure 6 and added a second graph magnifying the relevant range. Unfortunately, the data from Chen et al. (2018) is not published but visually falls within the range of the data already shown as grey background.

[Figure]

Referee 2:

The authors have solved all the issues I had with the earlier version of the manuscript. One new, but very minor issue arose from changes made to Figure 2. The line between "top copper base" and the cooling block currently points to the "vapor chamber", whereas it should point to the copper part above it.

Thank you very much, we have updated Figure 2 accordingly.